# Humoral and Cellular Immune Response in Asymptomatic Dogs with Visceral Leishmaniasis: A Review

**DOI:** 10.3390/vaccines10060947

**Published:** 2022-06-14

**Authors:** Ana García-Castro, Adriana Egui, María Carmen Thomas, Manuel Carlos López

**Affiliations:** Instituto de Parasitología y Biomedicina López Neyra, Consejo Superior de Investigaciones Científicas, 18016 Granada, Spain; agarcas@ipb.csic.es (A.G.-C.); adrianaegui@ipb.csic.es (A.E.); mcthomas@ipb.csic.es (M.C.T.)

**Keywords:** canine visceral leishmaniasis, asymptomatic disease, immune response, humoral response, Th1/Th2 response, lymphoproliferative response, immune biomarkers

## Abstract

Visceral leishmaniasis is one of the deadliest parasitic diseases in the world and affects both humans and dogs. The host immune response to *Leishmania* infection plays a critical role in the evolution of canine visceral leishmaniasis (CVL) and consequently in the manifestation of clinical signs. The asymptomatic form of the disease is a major concern in the diagnosis of CVL and in the transmission control of *Leishmania* infection. Asymptomatic dogs are found in large proportions in endemic areas and are an unquantifiable source of infection. The present review analyzes the possible relationship between the activation of the antigen-specific immune response of the host and resistance or susceptibility to CVL. The review focuses on works that address the characterization of the humoral and cellular immune response profile, at both the functional and phenotypic levels, in infected dogs. Most studies relate the absence of clinical symptomatology to an increased proliferative response and a Th1 cytokine profile. Despite the numerous findings pointing to a differential immune response in asymptomatic dogs, the contradictory results reported in this review highlight the importance of establishing a precise clinical classification of the disease, performing more longitudinal studies, and including a higher number of animals in trials.

## 1. Introduction

Visceral leishmaniasis (VL), caused by the protozoa *Leishmania infantum* (syn. *L. chagasi*, in Latin America) and transmitted by the bite of female phlebotomine sand flies [1], is the most severe and progressive form of leishmaniasis [2]. This important zoonotic disease affects both humans and dogs in endemic areas of the Mediterranean basin, Asia, and Latin America, and it is emerging in North America [3,4]. VL is one of the deadliest parasitic diseases in the world, causing an estimated 20,000 to 40,000 human deaths and 0.2–0.4 million new cases each year [5]. Infected dogs, whose prevalence in the canine population may reach up to 80% in highly endemic areas, are the main reservoir of the parasite in urban zones and play a key role in the transmission cycle of *L. infantum* to humans [4]. Indeed, it has been observed how an increase in canine visceral leishmaniasis (CVL) cases precedes a rise in human cases [6]. This highlights the importance of CVL control strategies to manage the spread of the parasite in human and canine populations and the need for early and comprehensive diagnosis of dogs infected with *L. infantum* [7].

Effective control of CVL includes vector control, prevention, treatment, or culling of infected dogs [5]. However, currently, there are still barriers to overcome, such as the early screening of infected dogs. Likewise, since there is no scientific evidence that supports that seropositive dog culling could reduce the incidence of VL, this control measure should be revised or sufficiently argued [2,5]. Drug treatment of these infected dogs is also expensive, and although the available protocols can promote clinical cure, improve quality of life and life expectancy, and reduce the parasite load and infectiousness to sand fly vectors, parasitological cures are rarely achieved, and the rates of relapse are high [2,8]. Vaccination of these animals is one of the most promising tools for the effective control of this disease. There are currently three commercially available vaccines against CVL: Leish-Tec^®^ (Ceva Animal Health, Brazil), CaniLeish^®^ (Virbac Santé Animale, France), and Letifend^®^ (Laboratorios Leti, Spain) [9]. Despite the reductions in infectiveness and disease progression that these vaccines showed in some trials [1,10], their protective efficacy is still low [2]. Since the levels of protection offered by immunization alone are not considered satisfactory for preventing *L. infantum* infection, the commercially available vaccines themselves recommend simultaneous administration with topical insecticides [11]. Furthermore, the application of low–moderate efficacy vaccines in endemic areas could have a negative impact on the diagnosis and control of Leishmaniasis, since these vaccines, by only reducing the appearance or severity of clinical signs, may mask the disease in infected individuals, thus constituting an important reservoir of the parasite, which in turn could indirectly induce an increase in the incidence of infection [12,13,14]. On the other hand, currently available studies on licensed vaccines are considered insufficient and do not allow for comparative studies between them due to the lack of standardization in study design, methodological shortcomings, and substantial differences in the characteristics of the populations evaluated [11].

The precise diagnosis of CVL may also be complex, as not all infected animals develop clinical manifestations [7]. In 1988, Mancianti et al. classified CVL into three clinical forms on the basis of major features observed in seropositive infected dogs: asymptomatic dogs (AD), who do not show signs of the disease; oligosymptomatic dogs (OD), with a maximum of three clinical signs, including opaque bristles and/or localized alopecia and/or moderate loss of weight; and symptomatic dogs (SD), who show some or all severe signs of the disease, such as opaque bristles, severe loss of weight, onychogryphosis, cutaneous lesions, apathy, and keratoconjunctivitis [15]. However, classification of dogs based solely on physical examination without considering clinical pathological abnormalities or the possibility of undetectable organ dysfunction may be insufficient and misleading. On this basis, more recently, different authors recommend following LeishVet guidelines, which have been developed from exhaustive review of evidence-based studies, clinical expertise, and consensus of opinions derived from critical debates [16,17]. In addition, some asymptomatic dogs cannot be detected by conventional serological tests, while through direct methods, such as PCR, the percentage of these seronegative asymptomatic dogs is considerably high in endemic areas [18]. This fact implies a major problem, since asymptomatic dogs have shown to be highly competent to transmit the parasite to the vector [19]. This is of great relevance because these animals that do not show clinical signs of CVL constitute an undetectable source of infection [20,21]. Although several studies have reported that symptomatic dogs infected with *L. infantum* are highly infectious to their sand fly vector compared to oligosymptomatic and asymptomatic dogs [20,22,23,24,25,26,27], there is no clear relationship between the stage of clinical evolution and infectivity to sand flies [20,22,28,29]. In this context, it has been reported that other factors such as the nutritional status of the dogs, the virulence of the parasite strain, the vectorial capacity of the sandfly species, and the strains involved in transmission could also play a role in the intensity of transmission [30].

The evolution of *Leishmania* infection and its clinical manifestations are the result of the complex interactions between the host immune system and the parasite [31]. As in many diseases caused by protozoan parasites, infection is the first contact between the parasite and its host; the host may kill the parasite due to innate and/or acquired immunity, or the parasite may survive due to an efficient mechanism that evades the host response. If the parasite survives, an intriguing dynamic relationship between host and parasite may result; when in balance, the host becomes an ‘asymptomatic carrier’, and when out of balance, the result is disease [32]. Hereby, there is a correlation between the distinct clinical forms of CVL and certain immunopathological characteristics during the progression of the disease. In this context, it has been observed that while symptomatic dogs display a susceptible profile response, asymptomatic dogs exhibit a resistance pattern [33]. On this basis, the study of the immune response in asymptomatic dogs and biomarkers associated with resistance profiles might guide the development of more effective vaccines and more precise diagnostic/prognosis tests. The identification of biomarkers related to asymptomatic disease maintenance would also allow us to evaluate the immunoprotection induced by vaccination or treatment strategies [6,33,34]. Learning about the immune response and biomarkers associated with these dogs would widen the knowledge we have about CVL and the capacity to create better diagnostic tools to control the disease. Thus, the aim of this work was to review the systemic immune response of asymptomatic dogs by including studies focused on serum and/or peripheral blood and to discuss the biological meaning of these findings and the gaps that need to be addressed.

## 2. Materials and Methods

The literature search was performed in the Medline database (PubMed) in accordance with PRISMA guidelines and using determined keywords that are described in Appendix A. The search was limited to articles published from 1990 to 2022. After the initial search and review of the references contained in each of the articles, reviews were excluded, and original research articles, reports, comparative studies, and short communications were selected.

Articles were included in this review according to the following inclusion criteria:Studies performed in dogs.Studies that analyze humoral and/or cellular immune responses in serum and/or peripheral blood of asymptomatic dogs infected with *L. donovani*, *L. chagasi*, or *L. infantum*.Studies that include not only a group of asymptomatic dogs but also a group of healthy noninfected control dogs and/or symptomatic dogs with visceral leishmaniasis.

Conversely, articles were excluded when they fell within one or more of the following categories:Studies unrelated to visceral leishmaniasis.Studies that focused on human and nondog animal models.Studies in which dogs were vaccinated, immunized, or treated.Studies that did not focus on the immune response of dogs.Studies in which the immune response was analyzed in tissues.Studies in which the clinical classification was incorrect or inexistent.

A more exhaustive literature review focused on the immune response was conducted using the clusters of terms described in Appendix A. The articles were selected as described in Figure 1.

It is worth mentioning that when extracting the information from each individual study, we took into account the following variables to avoid any kind of bias: *Leishmania* species, type of infection (experimental or natural), fluid analyzed (serum or peripheral blood), number of dogs studied, geographical location, source of origin and inoculation route of parasites in case of experimental infection, diagnostic methods, clinical evaluation, techniques employed for cytokine and immunoglobulin detection, and type of antibody used for immunoglobulin detection (monoclonal or polyclonal antibody).

## 3. Results

### 3.1. Phenotypic Characterization of the Lymphocyte Population

Several studies have been conducted on the immunological profile of peripheral blood mononuclear cells (PBMCs) in the context of CVL (Table 1). In most of them, asymptomatic dogs (AD) showed increased concentrations of total T lymphocytes [6,18,35,36,37,38], CD4 and CD8 T cells [6,18,35,38,39], and B cells [18,35,38] in comparison with symptomatic dogs (SD). When a distinction was made between asymptomatic dogs with negative (AD-I) and positive serology (AD-II), some authors observed that both AD-I and AD-II had higher levels of CD5^+^ and CD4^+^ T cells, whereas CD8^+^ T cells and CD21^+^ B cells were found only high in AD-II, decreasing the CD4^+^/CD8^+^ ratio in seropositive dogs [18]. In contrast, other researchers reported decreased values of CD5^+^, CD4^+^, CD8^+^, and CD21^+^ cells in groups of seropositive dogs (AD-II and SD) when compared to seronegative dogs (control healthy dogs (CD) and AD-I), despite the occurrence of clinical signs of CVL, with higher CD4^+^/CD8^+^ ratios in AD-I and SD [6]. However, this relationship between clinical status and lymphocyte profile was not observed in all studies. Some authors detected higher percentages of CD4^+^ and CD8^+^ T cells in all infected groups of naturally infected dogs, regardless of their clinical classification [40], and other studies did not even find significant differences between noninfected and infected dogs [41,42]. With regard to B cells, although some authors also found no significant differences among clinical groups, they did report a higher population of regulatory (IgD^hi^) B cells in SD compared to CD and AD [43]. Concerning the role of regulatory T cells (Tregs), very few studies focused on their involvement in asymptomatic disease of *L. infantum*-infected dogs. The percentage of Tregs (Foxp3^+^CD4^+^), which play a key role in T-cell activation, was decreased in all infected dogs, while the expression of Foxp3 in CD8^+^ cells was barely detected in any group [40]. Likewise, although the activation status of lymphocytes in PBMCs has been little studied, asymptomatic disease has been associated with a higher activation status of circulating lymphocytes, as these animals showed higher expression of MHC-II and an increased CD45RB/CD45RA expression index in comparison to SD and CD [35]. However, another study observed this upregulation of MHC-II^+^ in lymphocytes from peripheral blood in both asymptomatic and symptomatic animals compared to CD, although this increase was only significant in SD [42]. Concerning other markers involved in the regulation of T-cell activation, some authors have also studied the coinhibitory molecule PD-1, which was found to be overexpressed in CD4^+^ and CD8^+^ T cells from SD and AD compared to CD, with higher expression in SD [44].

#### 3.1.1. Cytolytic Activity of PBMCs

Even though resistance to *Leishmania* infection is associated with the cytokine profile of PBMCs, as we will examine in depth in this review, other mechanisms of action could be involved in the protective effect of T cells, such as cytolysis [45]. The research conducted by Pinelli et al. showed that PBMCs from asymptomatic, but not from symptomatic, dogs lysed autologous *L. infantum*-infected macrophages and that cytotoxic CD8^+^ T and even CD4^+^ T cells in some animals were involved and essential for this cytolytic response [45].

#### 3.1.2. Nitric Oxide Production

One of the mechanisms by which macrophages exert their antileishmanial activity is through the production of nitric oxide (NO) [46]. Given the role of this molecule in parasite clearance, it is also necessary to address the relationship between NO production in PBMCs and clinical signs of CVL. Panaro et al. analyzed NO production by PBMC-derived macrophages infected in vitro by *L. infantum* at 4 and 8 months after diagnosis [46]. The authors observed that, while at the first follow-up (4 months), macrophages from symptomatic dogs released higher levels of NO, the opposite occurred 8 months after diagnosis, when NO production in asymptomatic dogs suffered a substantial increase that was not observed in symptomatic dogs [46]. The correlation between NO production and asymptomatic disease was previously reported in naturally infected dogs from endemic areas of Brazil [47]. Souza et al., in a recent manuscript, also described the presence of higher NO production in all infected dogs compared to noninfected controls and higher NO levels in asymptomatic *versus* symptomatic dogs [48].

#### 3.1.3. Lymphoproliferative Response of PBMCs

The capacity of PBMCs from asymptomatic dogs to proliferate upon exposure to *Leishmania*-specific antigens has been extensively discussed (Appendix A). Multiple studies have found an association between the protective immunity of asymptomatic dogs and a strong proliferative response of PBMCs to several *Leishmania* antigens: soluble *Leishmania* antigen (SLA), frozen and thawed (f/t) antigens, amastigote extracts (AM), the purified promastigote protein gp63, several recombinant cysteine proteinases from *L. infantum* (rCPA, rCPB, CTE) and *L. chagasi* (rLdccys), amastigote protein P-8, and *L. infantum* recombinant antigens HSP-70, PFR-2, KMP-11, LeIF, and Ldp23 [44,45,47,49,50,51,52,53,54,55,56,57,58,59,60]. SLA is one of the most commonly used antigens in the lymphoproliferative assays. Figure 2 shows a comparative analysis of the lymphoproliferative response after SLA stimulus in AD and SD dogs. As observed in Figure 2, and even though in several studies the SLA stimulus induced a higher specific proliferative response in PBMCs from AD [50,57,58], in some studies, other authors observed similar lymphoproliferative responses to this antigen in both AD and SD [53,54,57,61,62]. PBMCs from AD, which did not proliferate upon stimulation with SLA, were shown to proliferate after exposure to other recombinant antigens, such as LeIF and Ldp23 [49]. PBMCs from experimentally infected AD also proliferate upon induction by HSP-70, PFR-2, or KMP-11 recombinant proteins, although at a lower level than that induced by SLA [54]. Moreover, the f/t lysate produced stronger proliferation in PBMCs from AD than the recombinant antigens rCPA and rCPB and the synthetic peptide CTE [51,52] but slightly lower proliferation than gp63 [60]. Likewise, *L. chagasi* cysteine proteinase Ldccys induced a higher PBMC proliferative response in AD than that obtained with amastigote extracts [47]. As expected, most of the reports showed that the response of PBMCs to mitogens (polyclonal activators), such as Con A, PHA, or PWM, was high and similar between infected and noninfected animals [51,52,53,54,55,57,59,60,61]. However, some authors have described a distinct mitogenic proliferative response between infected and noninfected animals [49]. This was the case of Carrillo et al., who reported a decreased proliferative response upon mitogens with disease progression [53].

#### 3.1.4. Cytokine Profile

The studies that analyzed the cytokine profile of PBMCs from asymptomatic dogs are listed in Appendix A. Resistance to *Leishmania* infection has been generally associated with a Th1 response characterized by the secretion of proinflammatory cytokines [45]. Thus, naturally and experimentally infected asymptomatic dogs have been shown to secrete high levels of IFN-γ [44,45,47,53,54,55,63,64,65], TNF-α [53,54,63], IL-2, and IL-18 [31,63]. In fact, some authors suggested that high expression of both IFN-γ and IL-2 in dogs who stay symptomless may indicate protection against disease progression [31]. Moreover, symptomatic dogs with severe CVL (clinical score higher than 7) showed the absence of IFN-γ, TNF-α, IL-2, IL-7, and IL-15 inflammatory mediators [65,66] and increased levels of IL-10, CCL2, and CXCL1 [65]. Thus, dogs with clinical signs of the disease exhibit a predominant Th2 response, characterized by elevated levels of the anti-inflammatory cytokines IL-4 and IL-10 [6,39,44,47,55,66,67,68], although significant levels of IL-4 have also been found by other authors in asymptomatic dogs that have been naturally [31,36] and experimentally infected [53,54,63]. Similar findings have been reported in AD concerning IL-10, whose expression has been detected in unstimulated PBMCs [63], whole blood [31], serum [65,69], and *Leishmania* antigen-stimulated PBMCs [43,62,63,64,67]. However, some studies described that specific antigen stimulation did not induce IL-10 expression in any group of infected dogs (AD and SD), suggesting that this cytokine may not have a predominant negative immunoregulatory role in CVL [53,54].

Yet, the relationship between Th1 cytokines and protection against *Leishmania* in CVL is not clear. Thus, it has also been reported that fresh PBMCs isolated from asymptomatic dogs experimentally infected with *L. infantum* showed expression of TNF-α, IL-2, IFN-γ, IL-10 and IL-18 mRNAs similar to those from noninfected dogs [31,63]. Likewise, some studies have called into question the role of IFN-γ alone as a resistance marker, as similar or higher levels of this cytokine were found in symptomatic dogs than in asymptomatic dogs [6,31,36,39,40,62,67,69]. Similarly, higher levels of IL-2 [31,36] and TNF-α [67] have been detected in SD, as well as IL-18, although an association of IL-18 with resistance or susceptibility could not be established [31,53,54].

### 3.2. Analysis of the Humoral Immune Response

The humoral response in CVL and the immunoglobulin subclasses that predominate in asymptomatic or symptomatic forms of the disease have been quite controversial. One of the most discussed topics is the correlation between IgG subclasses (IgG1 and IgG2) and disease progression, whose literature has been reviewed in Appendix A. Thus, while some researchers observed a correlation between IgG1 and asymptomatic infection [70,71,72,73], others have proposed IgG1 as a susceptibility marker associated with the appearance of clinical signs [53,57,74,75,76,77,78,79,80]. While IgG2 has also been associated with immune protection in AD, presenting high levels in these dogs [38,52,78], some studies detected higher levels of IgG2 in symptomatic animals [25,50,53,61,73,80,81,82,83,84]. However, this dichotomy in the levels of IgG1 and IgG2 responses in symptomatic and asymptomatic dogs was not observed in other studies, in which no correlation was found between these subclasses and the clinical status of *Leishmania*-infected dogs [18,55,62,85,86,87,88]. Interestingly, a recent study suggests the possibility that both IgG1 and IgG2 subclasses were associated with immune-protective mechanisms against *Leishmania* infection [89].

The use of polyclonal or monoclonal antibodies that recognize the different IgG subclasses could be the reason for the published contradictory results regarding their association with susceptibility or resistance in CVL. Quinnell et al., using monoclonal antisera, reported significant increases in IgG1, IgG2, IgG3, and IgG4 in polysymptomatic dogs from an endemic area of Brazil [85]. Oliveira et al., however, observed increased levels of IgG1 and IgG4 in AD *versus* SD using monoclonal antibodies [71]. Studies carried out in endemic zones of Brazil and in Iowa (USA) showed higher levels of IgG, IgG1, and IgG2 in symptomatic naturally infected dogs using polyclonal antibodies [55,87]. This variability in the obtained results using both monoclonal and polyclonal antibodies was also described by Marcondes et al. [90]. These authors found no significant differences in the levels of IgG1 and IgG2 between AD and SD dogs when using polyclonal antibodies and instead observed higher levels of all IgG subclasses except IgG2 in SD when using monoclonal antibodies [90]. In addition, Travi et al. used polyclonal antisera and did not find any significant difference in the IgG1 and IgG2 levels from experimentally infected asymptomatic and symptomatic dogs [62]. Interestingly, given the inconsistencies in the results despite the type of antisera used, a group of researchers pointed out the use of whole promastigote extracts or SLA in ELISA as a source of nuclear and cytoplasmic components that could create nonspecific binding of IgG subclasses [88]. For that purpose, they tested both SLA and the recombinant proteins LACK and LeIF in dogs from endemic areas in Tunisia and found higher levels of IgG, IgG1, and IgG2 in SD dogs, regardless of the antigen used [88].

IgM, IgA, and IgE levels have also been discussed in the literature. Symptomatic dogs have shown higher expression of IgE [18,70,74,82], IgA [56,57,70], and IgM [56,57,73] than asymptomatic dogs, which implies the failure of these isotypes to provide immunoprotection against *L. infantum*. In contrast, other studies reported no correlation between IgE and IgM and clinical status [70,73,75,81] and showed high levels of IgA and IgM in both symptomatic and asymptomatic dogs [18,73].

## 4. Discussion

Infection caused by *Leishmania infantum* induces host defense reactions to infection involving effector mechanisms of the innate and acquired immune responses. Given the status of *L. infantum* as an obligate intracellular infectious agent, the cellular response, mainly mediated by T lymphocytes, plays a critical role in infection control [91]. Thus, these cells of the immune system recognize parasite antigens and promote the specific functions necessary for their elimination. Resistance to canine visceral leishmaniasis (CVL) seems to be associated with higher levels of total T lymphocytes (CD4^+^ and CD8^+^) and B cells [6,18,35,37,38,39], whereas susceptibility to the disease is related to a decreased number of these cells. CD8^+^ T cells have an important protective role during CVL, not only because of their ability to mount a protective Th1 response during the early stages of infection but also due to their cytolytic activity against *L. infantum*-infected macrophages [45]. In the asymptomatic stage (AD) of visceral leishmaniasis, CD4^+^ T cells also play a role by being able to lyse infected macrophages, although the relevance of this subtype of cytotoxic activity for visceral leishmaniasis in vivo is not yet well known [45]. Circulating lymphocytes of asymptomatic dogs also presented elevated expression of MHC-II and a higher CD45RB/CD45RA ratio [35,42]. CD45RB has previously been related to CD4^+^ and CD8^+^ T cells activated by protozoan infection [92], whereas CD45RA seems to be highly expressed by naïve canine T cells, helper T cells secreting IFN-γ, and a wide range of B cells [35]. These results suggest enhanced antigen presentation ability and the effective activation of T cells. The reduced presence of regulatory CD4^+^ T cells (Treg) found in infected dogs would also optimize T-cell activation and effector functions during infection [40], as these Treg cells can suppress the antiparasitic CD4^+^ T-cell response [93]. However, although low levels of Tregs in asymptomatic dogs would allow stronger control of parasite growth, the activation of this immunoregulatory mechanism is critical to protect tissues from damage caused by excessive inflammation [40,93]. The regulation of the immune response is also influenced by regulatory (IgD^hi^) B cells, which, unlike B cells, were found at low levels in AD dogs but were increased in symptomatic dogs. It has been described that during symptomatic infection by *Leishmania*, IgD^hi^ B cells produce IL-10 and suppress IFN-γ production in T cells through the PD-L1/PD-1 and IL-10 pathways, leading to the suppression of the T-cell response and cellular exhaustion in these dogs [43]. In fact, the inhibitory receptor PD-1 has been found to be expressed at higher levels in SD than in AD [44]. This receptor, which is involved in the negative regulation of T-cell activation, seems to partially mediate CD8^+^ and CD4^+^ T-cell exhaustion in CVL [44].

Protective immunity against *L. infantum* in dogs has also been associated with a strong lymphoproliferative response of PBMCs to *Leishmania* antigen [45]. Most of the studies analyzed in this review showed a higher proliferation of PBMCs from experimental and naturally infected asymptomatic dogs in response to several *L. infantum* antigens [44,45,47,49,50,51,52,53,54,55,56,59,60]. In contrast, PBMCs from symptomatic dogs failed to respond to these parasite antigens or showed lower cell proliferative responses. In asymptomatic dogs, it has been suggested that those that show a poor cellular response are more prone to progress in the disease than dogs with stronger cell-mediated immunity [56]. Likewise, the inability to mount an effective and specific proliferative response observed in SD dogs would be indicative of the immune suppression that has been reported in these animals [51]. Some authors have described that this unresponsiveness occurred only in later stages of the disease and that, in early infection, symptomatic dogs were able to develop a proliferative cellular response to leishmanial antigens [57,61,62], although these studies have been conducted in a low number of dogs.

Concerning the cytokine profile, asymptomatic dogs seem to have a predominant Th1 response. In most of the studies published, these AD dogs showed high levels of one or more of these proinflammatory cytokines: IFN-γ [44,45,47,53,54,55,63,64,65], TNF-α [53,54,63], IL-2 [31,63], IL-18 [63,65,66], IL-6 [65,66], IL-15, and IL-7 [65]. The IFN-γ, TNF-α, and IL-2 cytokines have been proven to activate macrophages to keep the parasite under control and to avoid its dissemination through the production of nitric oxide (NO) and reactive oxygen species (ROS) [64,65]. Conversely, the decreased NO levels in SD may be related to the inhibitory effect on signal transduction for iNOS and NO production induced by the anti-inflammatory cytokines released in the symptomatic active form of the disease, such as IL-4, IL-10, IL-13, or TGF-β [46]. In fact, both IL-4 and IL-10 have been shown to inhibit the expression of the enzyme iNOS2, downregulating macrophage activity and allowing the persistence of parasites in blood and their transmission [67]. IFN-γ and TNF-α participate in the regulation and activation of inducible nitric oxide synthase (iNOS), the enzyme responsible for NO production in macrophages [46]. NO expression may have a protective role in asymptomatic dogs, although this molecule cannot be considered a resistance marker since high levels of NO expression have been detected in both symptomatic and asymptomatic animals [46,48]. IL-2, in addition to being involved in macrophage activation, may also be implicated in decreasing the adverse effects of the inflammatory response, as this cytokine has been shown to regulate the production of immunoglobulins by B cells and the differentiation of regulatory T cells [36]. IL-18 expression has also been associated with resistance to canine leishmaniasis by inducing Th1-cell development, IFN-γ production, and the activation of T- and NK-cell cytotoxicity [63]. The cytokine IL-6 has been described to be involved in the regulation of IFN-γ receptor expression [94], and IL-7R seems to play a relevant role in T-cell survival [95]. The cytokine IL-15 has also been shown to participate in the control of CVL infection in asymptomatic dogs [65]. In fact, IL-15 has the ability, in association with IL-12, to activate a strong proliferative response, promoting a decrease in programmed cell death protein 1 (PD-1) expression in lymphocytes as well as increases in the expression levels of the cytokines IFN-γ and TNF-α [96,97].

The relationship between the cellular Th1/Th2 response in CVL and its association with resistance and susceptibility may not be as clear in dogs as in other species [74]. Thus, Th1 cytokines, usually related to asymptomatic infection, have been detected in symptomatic dogs [6,31,36,40,62,67], and Th2 cytokines, usually associated with clinical disease, have been detected in asymptomatic dogs [31,36,43,53,54,63,64,65]. The Th2-type cytokine IL-10, which is associated with the suppression of cytokine production by Th1 cells and consequently with the development of a Th2 immune response [31], has been described could be overexpressed in IFN-γ-producing dogs as a negative feedback mechanism to control proinflammatory cytokines and reduce their detrimental effects on dog health [64]. Likewise, the expression of proinflammatory cytokines such as IL-12, IFN-γ, and TNF-α in dogs with CVL has also been called into question, as some studies have also shown that they may be involved in disease progression [31,36,67].

In the context of the evolution of CVL in dogs and in addition to cell-mediated immunity, another important factor to be considered is the humoral response and, particularly, the correlation between IgG subclasses (IgG1 and IgG2) specific to *Leishmania* antigens. Numerous studies have pointed to increased levels of anti-*Leishmania* IgG1 as a determinant factor for the symptomatic evolution of the disease [53,57,74,75,76,77,78,79,80]. It has been postulated that the ability of IgG1 to activate complement could contribute to increased pathologic manifestation of CVL in dogs, as this immune mechanism mediates inflammatory reactions [77]. However, other authors found no association between the presence of symptomatology and high IgG1 levels. IgG2 isotype antibodies have been associated with immune protection mechanisms against *L. infantum* infection [38,52,78] and have also been correlated with clinical symptoms of CVL [25,50,53,61,70,72,73,80,81,82,83,84]. The study of anti-*Leishmania* IgG subclass antibody production in a cohort of naturally infected dogs showed that the levels of all IgG subclasses were strongly intercorrelated and particularly elevated in sick dogs in which the presence of the parasite was detected by PCR. Thus, these results suggest that the evolution of CVL may be associated with the upregulation of antigen-specific antibodies of all IgG subclasses, particularly IgG1, IgG3, and IgG4 [85].

This strong but nonprotective humoral response observed by several authors in symptomatic dogs [38,55,87,88] would also be expected, as one of the characteristics of symptomatic disease is polyclonal B-cell activation [55]. It should be noted, however, that most of the articles that correlated IgG subclasses to susceptibility or resistance to sickness used polyclonal antisera for antibody detection. In contrast, the studies in which a monoclonal panel of antibodies was employed did not report a polarized response of the IgG subclass in dogs but showed a general increase in all IgG isotypes with disease progression [85,86,90]. Although some authors related these contradictory results with the use of polyclonal or monoclonal anti-Igs, the presence of cross-reactive autoantibodies, largely produced in this disease, may also influence the degree of specificity of the *Leishmania*-antigenic preparations and the results obtained [88]. Thus, the repertoires of autoantibodies against extracts of HEp-2, ds-DNA, human albumin, and transferrin as autoantigens indicated that in AD dogs, there are higher levels of IgG1 autoantibodies and a higher seroprevalence than in SD dogs, in which there are lower levels and lower seroprevalences of total IgG and IgG2 [88]. Moreover, data from competitive HEp-2-ELISA using total leishmanial antigens as inhibitors showed that in AD, IgG1 antibodies are predominantly autoantibodies to self-antigens, whereas in SD, they are mainly cross-reactive (*Leishmania*/self-antigens) [88]. Regarding the role of other immunoglobulins in CVL, such as IgE, IgA, and IgM, most of the studies reported high levels in symptomatic dogs. IgE, which is considered a serum marker of Th2 in different parasite infections, seems to be correlated with the symptomatic stage of CLV. In fact, higher expression of IgG1 and IgE was only present in symptomatic animals. This correlation between the expression of IgG1 and IgE and the pathology of leishmaniasis points to their potential role as markers of active disease [74,82]. The profile of anti-*Leishmania* antibodies in different clinical forms of canine visceral leishmaniasis (CVL) in naturally infected dogs was studied by Freitas et al. [73], who showed that both asymptomatic and symptomatic dogs presented increased levels of total IgG, IgA, and IgE in addition to IgG1 and IgG2. Moreover, IgG2 and IgM presented positive correlations with the clinical signs of the disease, while total IgG, IgG1, and IgA showed negative correlations. The increase in IgE did not show a correlation with the clinical changes in infected dogs [73]. However, Reis et al. (2006) demonstrated a positive correlation of patterns of IgA with the clinical status of naturally infected animals [70]. Increased production of IgA, which is involved in mucosal immunity, was described in infected dogs showing symptomatology, suggesting that the worsening clinical condition in dogs has also been linked to elevated IgA levels [56,57]. Additionally, it suggests that dogs developing a high T-cell response are probably able to avoid the dissemination of the parasite to mucosal surfaces and, as a consequence, to produce low or background specific IgA levels [56,57]. Further studies are needed to investigate the relationship between specific IgA and parasite load, especially at mucosal sites. Furthermore, the discovery of IgA deposits in the kidneys of infected dogs has suggested that this immunoglobulin may contribute to the generation of glomerulonephritis associated with this disease [70]. Regarding IgM, although this immunoglobulin has been typically associated with the acute forms of infectious diseases, significant levels have been detected in the chronic phases of CVL [18,56,57,70,73]. It has also been described that serum levels of anti-*Leishmania* IgM from naturally infected dogs (AD and SD) remain, with no significant differences compared to those from the noninfected control group [73], although, despite this finding, the authors observed a positive correlation with respect to the association with symptomatology [73].

Although there are numerous findings that point to the existence of a different immune response in asymptomatic *versus* symptomatic dogs, there are relevant discrepancies in the results obtained in the different studies carried out in this regard. Thus, it is essential to continue the research in this context and to take into account variables such as the breed of the dog, the genotype of the infectious agent, the coexistence with other pathologies, the type of infection (natural or experimental), as well as the sensitivity of the techniques used to evaluate the antigen-specific immune response induced by the parasite infection. Likewise, we also consider it necessary to carry out more longitudinal studies in infected dogs in order to evaluate the kinetics of the immune response throughout the infection and its association with the control of the pathology. Altogether, it will make it possible to find clear patterns capable of predicting the outcome of the infection and that are useful as biomarkers of evolution and as activation molecules for the design of therapeutic and/or preventive vaccines. In this context, we believe that the identification of biomarkers associated with asymptomatic or symptomatic CVL would allow us to monitor the efficacy of therapies and vaccines and to develop better diagnostic tests.

It is also relevant to highlight that most of the reported studies establish the clinical stage of the infected dog based solely on physical examination. This is of limited value as dogs without apparent clinical manifestations may be classified as asymptomatic despite having relevant alterations in serum and urinary biochemical parameters and/or some organ dysfunction [16,17,98]. Another aspect to consider is that most of the studies only included asymptomatic dogs with positive serological tests, excluding infected animals with low titers of anti-*Leishmania* antibodies, whose infection status can only be detected by PCR [18]. Some studies have described that asymptomatic dogs with positive or negative serology showed a differential humoral and cellular responses to *Leishmania* antigens [6,18]. Thus, the inclusion of the asymptomatic dogs with low titers of antibodies would be critical for understanding both the complex immune response triggered by infection and the factors involved in the symptomatology progression of canine visceral leishmaniasis. Overall, all these findings reinforce the idea that CVL is a complex multifactorial disease that is affected by a set of factors that are correlated and should not be evaluated in an isolated manner.

## Figures and Tables

**Figure 1 vaccines-10-00947-f001:**
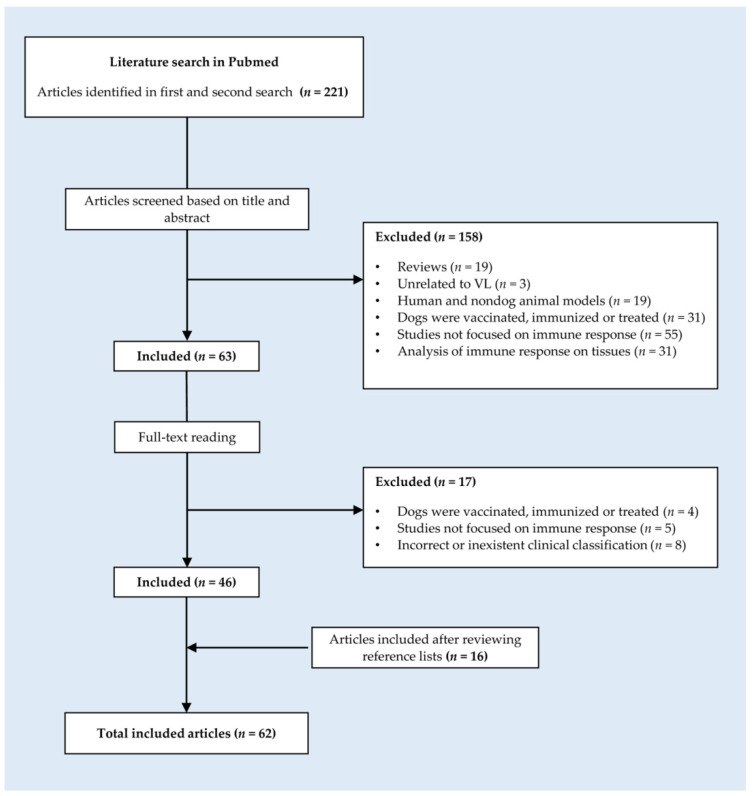
Flow diagram showing the steps followed to select the articles included in this review.

**Figure 2 vaccines-10-00947-f002:**
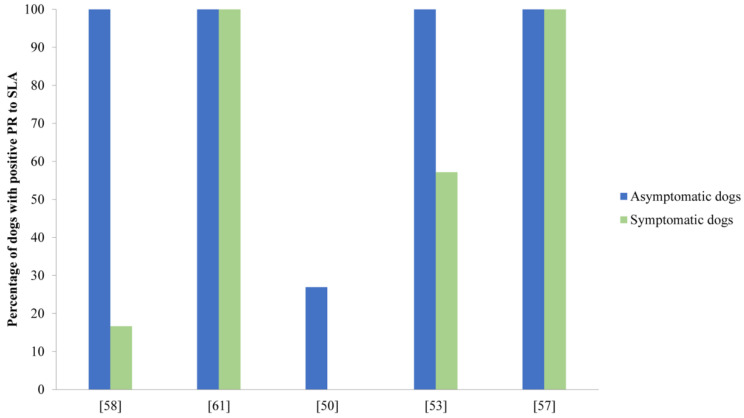
Graphical representation of the percentage of dogs who exhibited positive lymphoproliferative response to SLA. In this figure, all studies that specified the number of dogs who gave a positive proliferative response have been included. PR: proliferative response; SLA: soluble *Leishmania* antigen.

**Table 1 vaccines-10-00947-t001:** Studies that analyzed the immune profiles of PBMCs from infected and control dogs.

Cell Type Analyzed	Common Markers Studied	Additional Markers Studied	Clinical Classification (Number of Dogs	Geographical Location	Type of Infection	Methods	Main Findings	Reference
**T and B lymphocytes**	CD3/CD5, CD4, CD8	Foxp3	AD (*n* = 23 SD (*n* = 22) CD (*n* = 30)	Campania region (Italy)	Natural	IFA and FC with mAb	No differences between AD and SD for any marker. ↓% CD4^+^CD3^+^ cells and ↑ % CD8^+^CD3^+^ cells in AD and SD *versus* CD (****). ↓ Treg CD4^+^ cells in AD and SD *versus* CD (****).	[40]
		MHC-II	AD (*n* = 10) SD (*n* = 10) TD (*n* = 10) CD (*n* = 10)	Unknown	Natural	FC	↓ CD3^+^ T cells in SD *versus* AD and CD (*). Similar numbers of CD4^+^ and CD8^+^ T cells between groups. ↑ % MHC-II^+^ lymphocytes in SD *versus* CD (*).	[42]
		CD21, MHC-II, CD45RA, CD45RB	AD (*n* = 12) OD (*n* = 12) SD (*n* = 16) CD (*n* = 20)	Belo Horizonte (Brazil)	Natural	FC with mAb	↑ CD5^+^, CD4^+^ and CD8^+^ cells in AD *versus* SD (**). ↓ CD21^+^ cells in SD *versus* AD (**) and CD (*). ↑ MHC-II expression and CD45RB/CD45RA ratio in lymphocytes from AD *versus* SD, OD, and CD (*).	[35]
		CD21	AD (*n* = 6) SD/TSD (*n* = 8) CD (*n* = 22)	France	Natural	FC with mAb	↓ CD5^+^, CD4^+^, CD8^+^ and CD21^+^ cells in SD *versus* AD. ↓ CD21^+^ in AD *versus* CD (****).	[38]
			AD-I (*n* = 8) AD-II (*n* = 10) SD (*n* = 16) CD (*n* = 7)	Belo Horizonte (Brazil)	Natural	FC with mAb	↑ CD5^+^ and CD4^+^ cells in AD-I and AD-II *versus* SD (*). ↑ CD8^+^ cells in AD-II *versus* SD and CD (*). ↑ CD21^+^ cells in AD-II and CD *versus* SD (*).	[18]
			AD-I (*n* = 34) AD-II (*n* = 20) OD (*n* = 8) SD (*n* = 42) CD (*n* = 28)	Belo Horizonte (Brazil)	Natural	FC with mAb	↓ CD5^+^, CD4^+^, CD8^+^ and CD21^+^ cells in AD-II and SD *versus* AD-I and CD (****).	[6]
**T lymphocytes**	CD4, CD8	-	AD (*n* = 4) SD (*n* = 8) CD (*n* = 2)	Virginia (USA)	Experimental	FC with mAb	Similar numbers of CD4^+^ and CD8^+^ T cells between groups.	[41]
			AD (*n* = 20) SD (*n* = 20) CD (*n* = 20)	Atenas (Greece)	Natural	FC with mAb	↑ CD4^+^ T cells in AD and CD *versus* SD (***).	[39]
**B lymphocytes**	CD19, CD21, IgD, IgM	-	AD (*n* = 7) SD (*n* = 7) CD (*n* = 7)	USA and Natal (Brazil)	Natural	FC and FACS	Similar % CD19^+^ and CD21^+^ cells between groups. ↑ IgD^hi^ B cells in SD *versus* AD and CD (***).	[43]

IFA: immunofluorescence assay; FC: flow cytometry; FACS: fluorescence-activated cell sorting; PBMCs: peripheral blood mononuclear cells; mAb: monoclonal antibodies; AD: asymptomatic dogs; AD-I; asymptomatic dogs with negative serology; AD-II: asymptomatic dogs with positive serology; OD: oligosymptomatic dogs; SD: symptomatic dogs; TSD: treated symptomatic dogs; TD: treated infected dogs; CD: control healthy dogs; Treg: T regulatory cells. *p* < 0.05 (*), *p* < 0.01 (**), *p* < 0.001 (***), and *p* < 0.0001 (****).

## Data Availability

No new data were created or analyzed in this study. Data sharing is not applicable to this article.

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
