# Peer review of "Humoral and Cellular Immune Response in Asymptomatic Dogs with Visceral Leishmaniasis: A Review"

_vaccines, 2022, doi:10.3390/vaccines10060947_

Round 1

Reviewer 1 Report

The amount of time and effort put in by the authors to review the literature in the area of immune response to the asymptomatic infection in canine VL is appreciated. Information summarized in Tables 3, S1 and S2 may be useful, but is rather complicated, reflective of the complexity and uncertainty of the subject under review. One can come up with a number of issues or questions for author's consideration:

  1. Is asymptomatic infection true for all pathogens ? If so, shouldn't this be the starting point for the Introduction ? Are there any general or specific reviews on this subject ? If not, there may be inherent problems to clearly define symptomatic vs asymptomatic and infection vs no-infection, as discussed by the authors. Worthy of attention may be some discussion of such issues in recent publications related to Covid-19.
  2. Literature search in the Materials & Methods (Tables 1-2, Fig. 1) and Results section are presented with unnecessary details. It is at authors' discretion to select and include articles of relevance. This is taken for granted in a review and goes without lengthy statements in these sections.
  3.  The results or Findings presented are descriptive and diffused, listing mainly contradictory findings of immunological studies in different labs and locations, including the use of different reagents (polyclonal vs monoclonal 2nd antibodies). This also true for the bulk of the Discussion section. As a result, there is not much of new insight or concrete proposal to resolve the discrepancies in the last paragraph.
  4. The most positive statement was made in the Abstract for a need of longitudinal studies by following the infection and immunity in a given cohort for a long period. This is obviously easier to say than done. Important, but not mentioned, are other potential variables, e.g.  differences in dogs (different breeds) & infective agents (L. infantum/chagasi genotypes), different vector (sand fly species) factors, co-infection of the dogs with other parasites, etc.  

Reviewer 2 Report

This is a useful review of the published literature, though as the authors note, the results and methodology used by the different studies are variable. This makes it difficult to draw strong overall conclusions. The review is necessarily very descriptive, but I felt that the authors could have discussed more whether there were differences in results between different types of study eg experimental vs natural infections, longitudinal vs cross-sectional studies etc. For the few longitudinal studies, it would be useful to highlight how immune responses evolve through time.

My specific comments are as follows:

The results are described well and in a detailed way, but are not very ‘user-friendly’. The tables in particular are very detailed and it is hard to see what the overall patterns are. It should be possible to present some of the results graphically, eg the proportion of AD vs SD mounting a proliferative response in different studies, which would complement the tables and more clearly show the variation in results between studies.

Lines 69-72: dogs with asymptomatic disease are an important source of infection for sandflies, but results of studies comparing asymptomatic and symptomatic dogs are very variable, and there is no consensus that asymptomatic dogs are as infective as symptomatic. In addition to the cited refs, there are a number of studies that show a greater proportion of symptomatic dogs to be infectious.

Line 93: it is not clear why the systematic review was limited to articles written in English. As many studies have been performed in South America, why not include studies in Spanish and Portuguese?

Line 113: it also not clear why studies assessing immune responses in tissues were not included, as tissue immune responses may be as or more relevant that those in peripheral blood. I would recommend to include these.

Line 296: USA not EEUU

Lines 495-496: the paper ends quite suddenly by giving the conclusions of a single study. This study is not well linked to the previous text, and introduces two biomarkers (PGE2 and the antibody response to saliva) that have not been mentioned before. This needs a bit more discussion/integration.

Reviewer 3 Report

This is comprehensive review article on immune responses in dogs with Visceral Leishmaniasis (CVL). Asymptotic infected dogs act as the reservoir for Leishmania parasite, thus creating a major concern regarding the control of Leishmaniasis in humans in endemic area. Authors performed a thorough literature survey regarding the immune responses in asymptotic and symptomatic CVL and presented a comparative view on this subject. The review is well-constructed, and information enlisted are helpful for interested researcher. A few points could be further improved. Authors mentioned about three available vaccines for CVL, however, not many details have been provided. Additional information would be helpful to understand why these vaccines are not effective as expected. Most of the grammatical mistakes have been corrected in this version, besides few typos (mentioned below). In addition, it is not clear why this review article is formatted like a research article.

Line 165: MCH II should be MHC II. Please define.

“Analyzed”, some places it is written as analysed

Author Response

Reviewer 3 (Comments for the Author):

This is comprehensive review article on immune responses in dogs with Visceral Leishmaniasis (CVL). Asymptotic infected dogs act as the reservoir for Leishmania parasite, thus creating a major concern regarding the control of Leishmaniasis in humans in endemic area. Authors performed a thorough literature survey regarding the immune responses in asymptotic and symptomatic CVL and presented a comparative view on this subject. The review is well-constructed, and information enlisted are helpful for interested researcher. A few points could be further improved. Authors mentioned about three available vaccines for CVL, however, not many details have been provided. Additional information would be helpful to understand why these vaccines are not effective as expected. Most of the grammatical mistakes have been corrected in this version, besides few typos (mentioned below). In addition, it is not clear why this review article is formatted like a research article:

Following the reviewer's suggestion, a new paragraph including additional information on the three commercially available vaccines has been added to the end of the second paragraph of the introduction section. It is mentioned that the levels of protection offered by immunization alone with these vaccines are not considered satisfactory for preventing L. infantum infection and therefore the commercially available vaccines themselves recommend simultaneous administration with topical insecticides. It is also indicated that the studies currently available on licensed vaccines are considered insufficient and do not allow for comparative studies between them due to the lack of standardization in study design, methodological shortcomings and substantial differences in the characteristics of the populations evaluated. Thus, in the revised version of the manuscript, 4 new references have been included (11-14) in which all this information is provided and commented.

Regarding the format of the article, we have followed the PRISMA guidelines to prepare a structured review with a format similar to that of a systematic review.

Line 165: MCH II should be MHC II. Please define.

As indicated by the Referee, MCH-II has been replaced by MHC-II. In addition, to standardize the nomenclature throughout the text, MHCII has been replaced by MHC-II.

“Analyzed”, some places it is written as analysed.

Analysed has been replaced by analyzed in Tables S3, S4 and S5 (in the last row of the "main findings" column of Table S4, in the title of columns 1 and 2 of Table S5 and also in the title of tables S3, S4 and S5).

Reviewer 4 Report

The authors have revised this manuscript as required.

Author Response

Thank you.

Sincerely, MCL